# Global Health Priority Box: Discovering Flucofuron as a Promising Antikinetoplastid Compound

**DOI:** 10.3390/ph17050554

**Published:** 2024-04-25

**Authors:** Carlos J. Bethencourt-Estrella, Atteneri López-Arencibia, Jacob Lorenzo-Morales, José E. Piñero

**Affiliations:** 1Instituto Universitario de Enfermedades Tropicales y Salud Pública de Canarias, Universidad de La Laguna, Avda. Astrofísico Fco. Sánchez, S/N, 38203 La Laguna, Spain; atlopez@ull.edu.es; 2Departamento de Obstetricia y Ginecología, Pediatría, Medicina Preventiva y Salud Pública, Toxicología, Medicina Legal y Forense y Parasitología, Universidad de La Laguna, 38203 La Laguna, Spain; 3CIBER de Enfermedades Infecciosas (CIBERINFEC), Instituto de Salud Carlos III, 28029 Madrid, Spain

**Keywords:** chemotherapy, toxicity, programmed cell death, leishmaniasis, Chagas, flucofuron

## Abstract

Leishmaniasis, produced by *Leishmania* spp., and Chagas disease, produced by *Trypanosoma cruzi*, affect millions of people around the world. The treatments for these pathologies are not entirely effective and produce some side effects. For these reasons, it is necessary to develop new therapies that are more active and less toxic for patients. Some initiatives, such as the one carried out by the Medicines for Malaria Venture, allow for the screening of a large number of compounds of different origins to find alternatives to the lack of trypanocide treatments. In this work, 240 compounds were tested from the Global Health Priority Box (80 compounds with confirmed activity against drug-resistant malaria, 80 compounds for screening against neglected and zoonotic diseases and diseases at risk of drug resistance, and 80 compounds with activity against various vector species) against *Trypanosoma cruzi* and *Leishmania amazonensis*. Flucofuron, a compound with activity against vectors and with previous activity reported against *Staphylococcus* spp. and *Schistosoma* spp., demonstrates activity against *L. amazonensis* and *T. cruzi* and produces programmed cell death in the parasites. Flucofuron seems to be a good candidate for continuing study and proving its use as a trypanocidal agent.

## 1. Introduction

Diseases caused by kinetoplastids include three serious pathologies in humans, leishmaniasis, Chagas disease, and sleeping sickness, considered by the World Health Organization (WHO) as tropical neglected diseases (NTDs) [1]. Leishmaniasis and Chagas disease represent two of the most prevalent and debilitating neglected parasitic diseases in many tropical and subtropical regions of the world. These diseases, caused by protozoa *Leishmania* spp. and *Trypanosoma cruzi*, respectively, affect millions of people, with a significant impact on the quality of life of patients and public health.

Chagas disease is estimated to affect approximately 7 million people, mainly in Latin America, and can lead to major chronic diseases of the intestine and heart, which can kill up to 7000 people a year [2,3]. On the other hand, leishmaniasis affects more than 12 million people, causing, mainly in its visceral and most severe form, about 30 thousand deaths per year [4].

Standard treatments for Chagas disease include benznidazole and nifurtimox [5]. On the other hand, for leishmaniasis, miltefosine, pentavalent antimonials, amphotericin B, and paramomycin are the most used treatments [6]. Despite the therapeutic efforts of the last 50 years, current treatments for these diseases have important limitations, such as toxicity, the development of resistance, and prolonged treatment. In addition, some of the available drugs are not fully effective in all stages of the disease or all strains of the parasites [5,7].

In this context, the search for new therapeutic compounds has become an urgent priority. The last few decades have seen a major impact on this search for new therapies for leishmaniasis and Chagas disease. This research has mainly focused on pharmacological repurposing or the development of new compounds, both of natural and synthetic origin, in addition to other more novel lines such as nanotechnology [8,9] or immunotherapy [10]. The compounds of natural origin discovered in this period include lignans from *Piper jericoense* [11], sesquiterpenoids and flavonoids from *Inula viscosa* [12], indolocarbazoles from *Streptomyces sanyensis* [13], celastroloids from *Maytenus chiapensis* [14], or extracts from microalgae, such as *Chlorella vulgaris* [15]. On the other hand, chemotherapy of synthetic origin that has shown activity against *Trypanosoma cruzi* and *Leishmania* spp. during this period includes, among others, nitrotriazoles [16], quinolines [17], bacteriocins of *Enterococcus faecalis* [18], acrylonitriles [19], or bisabolol derivatives [20].

Innovation in the identification and development of molecules with activity against these parasites offers hope for improving current therapeutic regimens. In this regard, the Medicines for Malaria Venture (MMV, Switzerland) has been working since 1999 on researching and improving malaria treatments. One of its key initiatives is based on libraries or “boxes” of compounds containing hundreds of different molecules with potential antimalarial activity. These boxes are accessible to researchers, and MMV distributes them free of charge, thus accelerating the discovery of new treatments for malaria and other pathologies [21]. An example of such libraries is the Global Health Priority Box, which contains 240 compounds with activity against malaria and vectors or compounds designed to test against zoonotic diseases [22].

Due to their demonstrated activity, the 240 compounds included in the Global Health Priority Box were tested against *L. amazonensis* and *Trypanosoma cruzi*, as well as against macrophages to elucidate their selectivity. For the most selective compounds, different events related to their mechanism of action were performed to demonstrate the type of cell death they produce in the parasites.

## 2. Results

### 2.1. Activity against Trypanosoma cruzi and Leishmania amazonensis

The activity of the compounds included in the Global Health Box was studied with screening against the extracellular form of both kinetoplastids, *Leishmania amazonensis,* and *Trypanosoma cruzi*. The results of these 240 compounds were expressed as percentages of inhibition of the parasite populations relative to the negative control for *L. amazonensis* (Figure 1) and *T. cruzi* (Figure 2).

The 240 compounds of the box were divided into groups of activity, very low (<30%), low (30–50%), medium (50–70%), high (70–90%), and very high (>90%), included in Figure 3 and Figure 4.

In addition, a statistical study of the different sets of compounds of the Global Health Priority Box was performed. The compounds were divided by percentage of activity, depending on the previous use. The results against *Leishmania amazonensis* were included in Figure 5, and the results of *Trypanosoma cruzi* were included in Figure 6.

The most active compound, flucofuron, was selected to determine the inhibitory concentration 50 (IC_50_) against the extracellular forms of both parasites, the promastigote stage of *L. amazonensis* (6.07 ± 1.11 µM) and the epimastigote stage of *T. cruzi* (4.28 ± 0.83 µM), and the intracellular forms (3.14 ± 0.39 µM against *L. amazonensis* and 3.26 ± 0.34 µM against *T. cruzi*).

### 2.2. Cytotoxicity against Murine Macrophages

To find out the possible toxicity of this compound, the cytotoxicity against murine macrophages (CC_50_) was determined. Also, the relation between cytotoxicity and activity, the selectivity index (SI), was verified and included in Table 1.

The results show that flucofuron, against *L. amazonenesis*, has similar values of activity to the standard drug, miltefosine, as well as similar selectivity indices. Against *T. cruzi*, flucofuron demonstrates similar activities to benznidazole, but the selectivity indices are quite different due to the difference in cytotoxicity between the two compounds.

### 2.3. Mechanisms of Cell Death

To determine the type of cell death triggered by flucofuron in the parasites, different events relative to programmed cell death were studied.

#### 2.3.1. Chromatin Condensation Analysis

To determine if the chromatin was condensed, the Vybrant™ Apoptosis Assay Kit (ThermoFisher Scientific, Waltham, MA, USA) was used. The incubation of the promastigotes of *L. amazonensis* and the epimastigotes of *T. cruzi* with the IC_90_ of flucofuron and Vybrant are shown in Figure 7 and Figure 8, respectively. Condensed chromatin was visualized in blue in the DAPI channel by the action of the Hoechst, and cell deaths appeared in red in the RFP channel by the action of propidium iodide (PI).

#### 2.3.2. Mitochondrial Membrane Potential Analysis

To analyze alterations of the mitochondrial membrane potential, the JC-1 Mitochondrial Membrane Potential Assay Kit^®^ (Cayman Chemical, Ann Arbor, MI, USA) was used. The percentages relative to the negative control of the mitochondrial membrane potential were included in Figure 9 and Figure 10.

#### 2.3.3. Analysis of ATP Levels

To determine the levels of ATP, the CellTiter-Glo^®^ Luminescent Cell Viability Assay kit (Promega, Madison, WI, USA) was used. Figure 11 and Figure 12 include the percentage of ATP relative to the negative control of *L. amazonensis* and *T. cruzi*, respectively, treated with flucofuron.

#### 2.3.4. Integrity of the Membrane Permeability Analysis

The alterations in the plasmatic membrane permeability of the parasites treated with flucofuron were included in Figure 13 and Figure 14. The cells that have the plasmatic membrane permeability altered could be seen with the green fluorescence of Sytox^®^ Green kit (ThermoFisher Scientific, Waltham, MA, USA) in the GFP channel.

#### 2.3.5. Analysis of the Presence of Reactive Oxygen Species

To demonstrate the presence of oxidative stress, the CellROX^®^ Deep Red Reagent (ThermoFisher Scientific, Waltham, MA, USA) was used. The presence of reactive oxygen species could be observed in the Cy5 channel with the fluorescence presented in Figure 15 and Figure 16.

## 3. Discussion

Some antimalarial compounds have shown activity against kinetoplastids. Previous studies have shown that chloroquine, hydroxychloroquine [23], and primaquine [24] have activity against different species of *Leishmania*. Moreover, chloroquine also showed improved activity when used together with benznidazole, the standard treatment against *Trypanosoma cruzi* [25]. On the other hand, insecticides used for vector elimination have also been tested against some parasites, sometimes obtaining activity against kinetoplastids [26], although this is a complex field due to parasite adaptations.

For these reasons, the Global Health Priority Box seems to be a good source of compounds that could have activity against *Leishmania amazonensis* and *Trypanosoma cruzi*.

Of the 240 compounds used against *L. amazonensis*, 62 demonstrated the inhibition of more than 50% of the population, with particular attention on the 18 that produced more than 90% inhibition. In the case of *T. cruzi*, 98 compounds inhibited more than 50% of the parasite population, but none inhibited more than 90%.

Seeing the results divided by previous use against *L. amazonensis*, ZND and MB2 plates had a similar quantity of compounds, with inhibition of more than 50% (26 and 27 compounds, respectively). The VEC plate adds up to only nine compounds in this group. In the case of *T. cruzi*, the opposite was true, with plates ZND and MB2 containing fewer compounds with inhibition greater than 50% (28 and 24 respectively), while plate VEC contained 46 compounds with inhibition greater than 50%.

Against *L. amazonensis* and *T. cruzi*, flucofuron demonstrated a higher percentage of inhibition; for this reason, flucofuron was selected to study the activity against intracellular and extracellular forms of both parasites. It is important to note that in the case of *L. amazonensis*, flucofuron showed greater selectivity than the standard treatment, miltefosine, against both parasitic forms (SI = 13.8 in promastigote form and SI = 26.7 in amastigote form).

One of the points to be taken into account in the search for new compounds is the mechanism of action they follow to inhibit the parasites. The aim is to demonstrate the presence of events related to programmed or apoptotic cell death in the parasites, avoiding compounds that produce necrotic cell death, as this would produce an undesirable inflammatory response in the hosts [27]. Some events have been previously relative to programmed cell death, among these, the accumulation of reactive oxygen species (ROS), alterations in mitochondrial membrane potential, exposure to phosphatidylserine, decrease in cellular ATP levels, disruption in the plasma membrane, DNA fragmentation, and chromatin condensation [27,28,29,30]. In the present work, some of these events were studied to elucidate the type of cell death produced by flucofuron in parasites. Observing the results of the mechanisms of action, flucofuron induced chromatin condensation, reactive oxygen species accumulation, and plasma membrane permeability in parasites, as well as alterations in ATP levels and mitochondrial membrane potential. All these events demonstrated that flucofuron produces programmed cell death in both parasites.

Finally, we performed an in silico prediction of the ADME-Tox properties of flucofuron, and the results showed that this compound has good druglikeness. Firstly, flucofuron, despite not having a high gastrointestinal absorption, is not able to cross the blood–brain barrier, which would avoid possible side effects at the central nervous system level in patients. Furthermore, it is not predicted to be a substrate of the P-gp protein, so it would not be expelled from the cells into which the drug penetrates. On the other hand, regarding its capacity as a future treatment to be administered orally, we see that it complies with Veber’s and Lipinski’s rules [31].

In summary, flucofuron showed activity against *Trypanosoma cruzi* and *Leishmania amazonensis,* consistent with previous studies showing activity of flucofuron against different microorganisms, including bacteria, such as *Staphylococcus* spp. [32], and even parasites, demonstrating activity against schistosomiasis [33].

## 4. Materials and Methods

### 4.1. Products

The compounds tested in this study were donated by the Medicines for Malaria Venture (MMV). These 240 compounds were included in the Global Health Priority Box and divided into three sets, one with 80 compounds with confirmed activity against drug-resistant malaria (MB2 plate), one with 80 compounds donated by Bristol–Myers Squibb for screening against neglected and zoonotic diseases and diseases at risk of drug resistance (ZND plate), and one with 80 compounds with activity against various vector species, selected with experts from the IVCC (Liverpool School of Tropical Medicine, Liverpool) (VEC plate) [22]. The compounds were solved in dimethyl sulfoxide (DMSO) at 10 mM and stored at −20 °C.

Flucofuron (4,4’-Dichloro-3,3’-bis(trifluoromethyl)carbanilide), presented in the Figure 17, was acquired from LGC Standards, Dr. Ehrenstorfer^TM^ (LGC group, Barcelona, Spain); this is a solid powder stored at 20 ± 4 °C. The flucofuron was also solved in DMSO at 20 mg/mL and once dissolved, it is stored at −20 °C.

### 4.2. Cultures

Epimastigote forms of *Trypanosoma cruzi* (Y strain) cultured at 26 °C in liver infusion tryptose (LIT) supplemented with 10% of FBS and promastigote forms of *Leishamania amazonensis* (MHOM/BR/77/LTB0016) cultured at 26 °C in Schneider’s medium (SND) (Sigma-Aldrich, Darmstadt, Germany) supplemented with 10% of fetal bovine serum (FBS) were used. For the cytotoxicity assays, murine macrophages (J774A.1) cultured at 37 °C with a 5% CO_2_ atmosphere in Dulbecco’s Modified Eagle Medium (DMEM) (ThermoFisher Scientific, Waltham, MA, USA) supplemented with 10% of FBS were used.

### 4.3. Antiparasitic Activity

To develop the trypanocidal activity of the Global Health Priority Box, an initial screening was developed against the extracellular forms of the parasites, the promastigote stage of *Leishmania amazonensis* and the epimastigote stage of *Trypanosoma cruzi*. For this experiment, a known concentration of parasite is added to 96-well plates (10^6^ parasites/mL) with a concentration of each compound (10 µM). To determine parasite growth, 10% of the alamarBlue Cell Viability Reagent^®^ (ThermoFisher Scientific, Waltham, MA, USA) was used. The results were expressed as the percentage of live cells relative to the negative control.

The most active compound in both parasites (flucofuron) was selected to develop the inhibitory concentration 50 (IC_50_) against extracellular forms of the parasites, a concentration which inhibits 50% of the parasite population. Serial dilutions of flucofuron were added to a 96-well plate with 10^6^ parasites/mL and 10% of alamarBlue. After 72 h of incubation, the fluorescence (544 nm excitation, 590 nm emission) of the plate was determined using the EnSpire Multimode Plate Reader^®^ (PerkinElmer, Thermo Fischer Scientific, Madrid, Spain). Finally, the IC_50_ was calculated using a nonlinear regression analysis in Graphpad Prism 9.0.0. [34].

The IC_50_ against the intracellular form of the parasites, the amastigote stage, was developed using the same colorimetric method based on alamarBlue. In 96-well plates, 10^5^ macrophages per well were added. After complete adherence of the cells to the well (at least 2 h), 10^6^ parasites in *L. amazonensis* and 5 × 10^5^ parasites in *T. cruzi* per well were added to reach 1:10 and 1:5 macrophage:parasite, respectively. A total of 24 h later, after the internalization of the parasites, the non-internalized parasites were removed, and serial dilutions of the flucofuron were added. After 24 h, the treatment was removed, and 30 µL of sodium dodecyl sulphate 0.05% (SDS) was added to facilitate the lysis of the macrophages; after 30 s, medium was added to reach 200 µL and stop the action of the SDS. Finally, 10% of alamarBlue was added, and after 72 h of incubation at 26 °C, the fluorescence was determined and the IC_50_ was calculated [35,36].

### 4.4. Cytotoxicity against Murine Macrophages

One of the main problems with current kinetoplastid treatments is the occurrence of adverse effects due to toxicity. For this reason, it is important to determine the in vitro toxicity of the compounds. The cytotoxic concentration 50 (CC_50_), the concentration that inhibits 50% of the cell population, against murine macrophages was determined. To perform the assay, in a 96-well plate, 10^4^ macrophages were added to each well with serial dilutions of flucofuron and 10% of alamarBlue. After 24 h of incubation at 37 °C, the fluorescence of each well was determined, and the CC_50_ was calculated using a nonlinear regression analysis in Graphpad Prism 9.0.0. [13].

The selectivity index (SI) reports the greater ability of compounds to kill parasites than to damage mammalian cells. The SI of flucofuron was calculated as the ratio CC_50_/IC_50_.

### 4.5. Mechanisms of Cell Death

In addition to its activity and toxicity, it is important to know what kind of cell death flucofuron produces in parasites. For this purpose, different commercial kits were used to observe the presence of certain events related to programmed cell death. The IC_90_ of the flucofuron was incubated with the extracellular stages of the parasites for 24 h, and the kits were added with the manufacturer’s indications.

### 4.6. Chromatin Condensation Analysis

To determine the presence of chromatin condensation, the Vybrant^®^ Apoptosis Assay Kit n◦5 (ThermoFisher Scientific, Waltham, MA, USA) was used. The IC_90_ of flucofuron was incubated with the parasites for 24 h; after that, centrifugation was carried out (3000 rpm, 10 min, 4 °C), and the treated cells were resuspended in 50 µL of buffer. The Hoechst (5 µg/mL) and the PI (1 µg/mL) were added, and after 20 min of incubation at 26 °C, some pictures were captured with the EVOS^®^ FL Cell Imaging System (ThermoFisher Scientific, Waltham, MA, USA) using a DAPI light cube for Hoechst 33342, which dyed the chromatin condensed blue (excitation 350 nm/emission 461 nm), and an RFP light cube for PI, which dyed the death cells red (excitation 535 nm/emission 617 nm) [37].

### 4.7. Mitochondrial Membrane Potential Analysis

Mitochondria are indispensable for cell survival, so a disruption in mitochondria can lead to cell death, especially in kinetoplastids that have only one mitochondrion. To see changes produced in the mitochondria membrane potential, the JC-1 Mitochondrial Membrane Potential Assay Kit^®^ (Cayman Chemical, Ann Arbor, MI, USA) was used. After treating the cells with the IC_90_ of flucofuron, the cells were centrifuged (3000 rpm, 10 min, 4 °C), resuspended in 50 µL of buffer, and transferred to a black 96-well plate. As per the manufacturer’s instructions, 5 µL of the kit was added to each well, and after 30 min of incubation, the fluorescence was measured using the EnSpire Multimode Plate Reader^®^ (PerkinElmer). The results were expressed as the mean ± standard deviation of the J-monomers (green fluorescence, excitation 540 nm/emission 470 nm) to J-aggregates (red fluorescence, excitation 485 nm/emission 535 nm) ratio performed in duplicate on three different days. A standard treatment (miltefosine for *L. amazonensis* and benznidazole for *T. cruzi*) and a positive control was added (carbonyl cyanide m-chlorophenyl hydrazone, CCCP, 100 μM for 3 h) [38].

### 4.8. Analysis of ATP Levels

ATP is an indicator of cell viability; a drop in energy may indicate that the cell is in the process of cell death. To determine the levels of ATP in the parasites, the CellTiter-Glo^®^ Luminescent Cell Viability Assay (Promega, Madison, WI, USA) was used. After treating the cells with the IC_90_ of flucofuron, the cells were centrifuged (3000 rpm, 10 min, 4 °C), resuspended in 25 µL of buffer, and transferred to a white 96-well plate. As per the manufacturer’s instructions, 25 µL of the kit was mixed well with the cells (2 min with intense agitation); after 10 min of incubation, the luminescence was measured using the EnSpire Multimode Plate Reader^®^ (PerkinElmer). Miltefosine for *L. amazonensis* and benznidazole for *T. cruzi* were used as standard treatments, and sodium azide (NaN_3_ 20 mM for 3 h) was used as a positive control. The results were expressed as mean ± standard deviation of the percentage relative to the negative control (non-treated cells), performed in duplicate on three different days [14].

### 4.9. Integrity of the Membrane Permeability Analysis

The plasma membrane is indispensable for the survival of the cell, as it keeps the cell isolated and protected from the outside. For this reason, failure of the plasma membrane may be indicative of a cell death process in the cell. To observe the integrity of the plasmatic membrane permeability, the SYTOX^®^ Green nucleic acid stain fluorescent dye (ThermoFisher Scientific, Waltham, MA, USA) was used. This contains a dye that can only penetrate the affected cells, emitting green fluorescence. To perform the assay, the treated cells were centrifuged (3000 rpm, 10 min, 4 °C) and resuspended in 50 µL of buffer. The kit was added at 1 µM, following the manufacturer’s instructions. After 15 min of incubation with the kit, fluorescence (GFP light cube, excitation 504 nm/emission 523 nm) pictures were taken with the EVOS^®^ FL Cell Imaging System (ThermoFisher Scientific, Waltham, MA, USA). Standard treatments were added, including benznidazole for *T. cruzi* and miltefosine for *L. amazonensis* and a positive control, triton 0.1% [39].

### 4.10. Analysis of the Presence of Reactive Oxygen Species

Oxidative stress occurs in cells that have lost or altered their ability to eliminate reactive oxygen species (ROS), a process that can be toxic and is associated with cell death. To determine the presence of reactive oxygen species, CellROX^®^ Deep Red Reagent (ThermoFisher Scientific, Waltham, MA, USA) was used. This kit contains a reagent that emits fluorescence when oxidized by ROS. To perform the assay, the parasites incubated with the IC_90_ of the compounds were centrifuged (3000 rpm, 10 min, 4 °C) and resuspended in 50 µL of buffer. The kit was incubated at 5 µM for 30 min, and the images were captured with the EVOS^®^ FL Cell Imaging System (ThermoFisher Scientific, Waltham, MA, USA) in the Cy5 light cube (excitation 644 nm/emission 665 nm). Standard treatments were added, including benznidazole for *T. cruzi* and miltefosine for *L. amazonensis* and a positive control, hydrogen peroxide (H_2_O_2_) 600 mM for 30 min [40].

### 4.11. Statistical Analysis

The IC_50_, IC_90_, and CC_50_ were determined in duplicate on three different days using non-linear regression [inhibitor] vs. response—variable slope (four parameters) using non-linear regression analysis with 95% confidence using the GraphPad Prism 9.0.0 statistical software (Appendix A). The analysis was developed using a Tukey’s test, considering significant values of *p* < 0.05. The ANOVA analysis was also performed in GraphPad Prism 9.0.0.

## 5. Conclusions

Compound libraries, such as the Global Health Priority Box, are a good source of compounds to test for kinetoplastid diseases. Flucofuron demonstrated activity against *Trypanosoma cruzi* (SI against epimastigote stage: 19.6; SI against promastigote stage: 25.7) and *Leishmania amazonensis* (SI against promastigote stage: 13.8; SI against promastigote stage: 26.7). In addition, events related to programmed cell death were positive in the parasites treated with flucofuron, showing chromatin condensation, membrane permeability alterations, a decrease in the ATP levels, the accumulation of reactive oxygen species, or mitochondrial membrane potential alterations. In conclusion, flucofuron was shown to produce programmed cell death in these parasites. For these reasons, flucofuron is a good candidate to continue studies to use it in the future as a treatment for leishmaniasis and Chagas disease.

## Figures and Tables

**Figure 1 pharmaceuticals-17-00554-f001:**
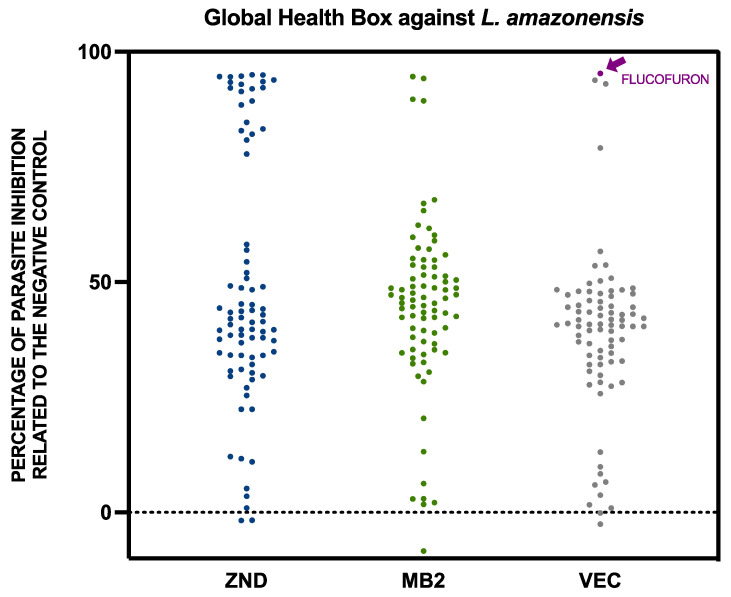
Activity against *Leishmania amazonensis* of the 240 compounds included in the Global Health Priority Box. The results were expressed as percentages of inhibition relative to the negative control. The flucofuron is indicated with an arrow in purple. Each point represents a compound included in the global health priority box, divided into three groups, including MB2, ZND, and VEC. MB2: 80 compounds with confirmed activity against drug-resistant malaria; ZND: 80 compounds donated by Bristol–Myers Squibb for screening against neglected and zoonotic diseases and diseases at risk of drug resistance; VEC: 80 compounds with activity against various vector species, selected with experts from the IVCC (Liverpool School of Tropical Medicine, Liverpool).

**Figure 2 pharmaceuticals-17-00554-f002:**
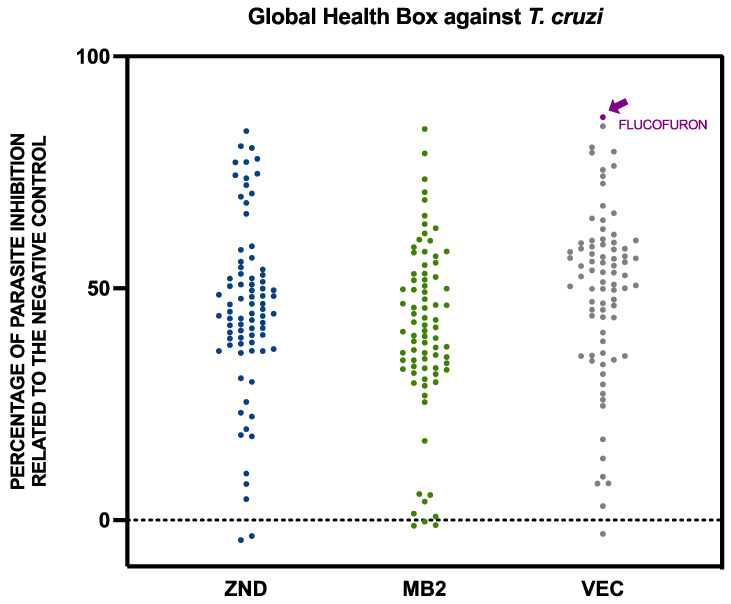
Activity against *Trypanosoma cruzi* of the 240 compounds included in the Global Health Priority Box. The results were expressed as percentages of inhibition relative to the negative control. The flucofuron is indicated with an arrow in purple. Each point represents a compound included in the global health priority box, divided into three groups, including MB2, ZND, and VEC. MB2: 80 compounds with confirmed activity against drug-resistant malaria; ZND: 80 compounds donated by Bristol–Myers Squibb for screening against neglected and zoonotic diseases and diseases at risk of drug resistance; VEC: 80 compounds with activity against various vector species, selected with experts from the IVCC (Liverpool School of Tropical Medicine, Liverpool).

**Figure 3 pharmaceuticals-17-00554-f003:**
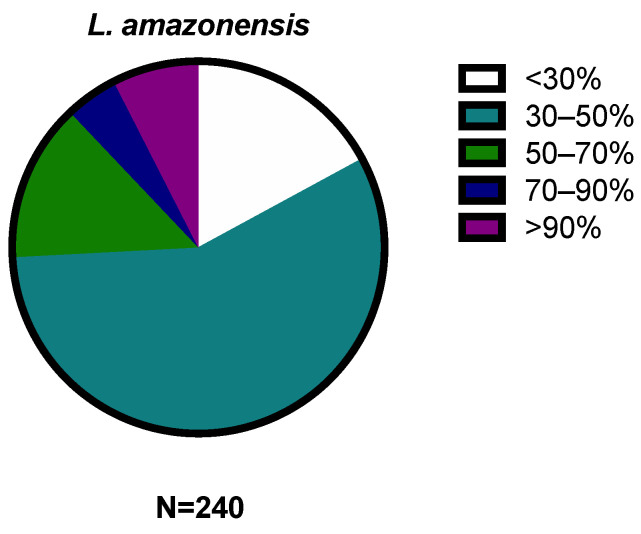
Distribution of results by percentage of inhibition against *Leishmania amazonensis*. The results were expressed as percentages relative to the negative control. (N = 240; <30% = 41; 30–50% = 137; 50–70% = 33; 70–90% = 11; >90% = 18.).

**Figure 4 pharmaceuticals-17-00554-f004:**
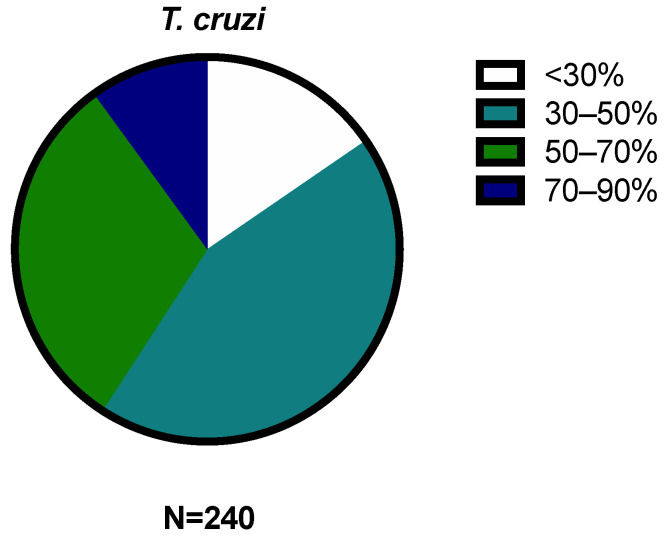
Distribution of results by percentage of inhibition against *Trypanosoma cruzi.* The results were expressed as percentages relative to the negative control. (N = 240; <30% = 37; 30–50% = 105; 50–70% = 74; 70–90% = 24; >90% = 0).

**Figure 5 pharmaceuticals-17-00554-f005:**
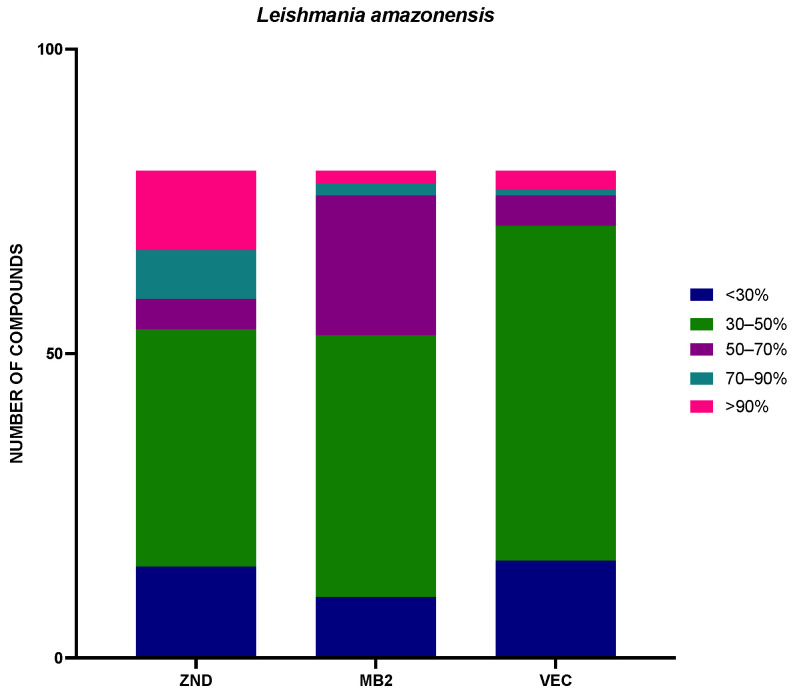
Distribution by plate of the percentage of inhibition against *Leishmania amazonensis*. The results were expressed as percentages relative to the negative control. MB2: 80 compounds with confirmed activity against drug-resistant malaria (MB2 = 80; <30% = 10; 30–50% = 43; 50–70% = 23; 70–90% = 2; >90% = 2); ZND: 80 compounds donated by Bristol–Myers Squibb for screening against neglected and zoonotic diseases and diseases at risk of drug resistance (ZND = 80; <30% = 15; 30–50% = 39; 50–70% = 5; 70–90% = 8; >90% = 13); VEC: 80 compounds with activity against various vector species, selected with experts from the IVCC (Liverpool School of Tropical Medicine, Liverpool) (VEC = 80; <30% = 16; 30–50% = 55; 50–70% = 5; 70–90% = 1; >90% = 3).

**Figure 6 pharmaceuticals-17-00554-f006:**
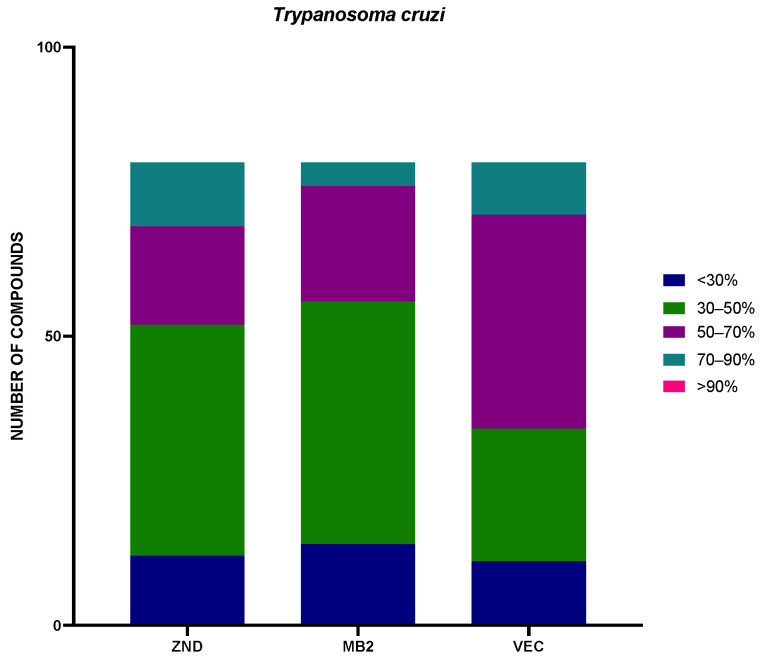
Distribution by plate of the percentage of inhibition against *Trypanosoma cruzi*. The results were expressed in percentage relative to the negative control. MB2: 80 compounds with confirmed activity against drug-resistant malaria (MB2 = 80; <30% = 14; 30–50% = 42; 50–70% = 20; 70–90% = 4; >90% = 0); ZND: 80 compounds donated by Bristol–Myers Squibb for screening against neglected and zoonotic diseases and diseases at risk of drug resistance (ZND = 80; <30% = 12; 30–50% = 40; 50–70% = 17; 70–90% = 11; >90% = 0); VEC: 80 compounds with activity against various vector species, selected with experts from the IVCC (Liverpool School of Tropical Medicine, Liverpool) (VEC = 80; <30% = 11; 30–50% = 23; 50–70% = 37; 70–90% = 9; >90% = 0).

**Figure 7 pharmaceuticals-17-00554-f007:**
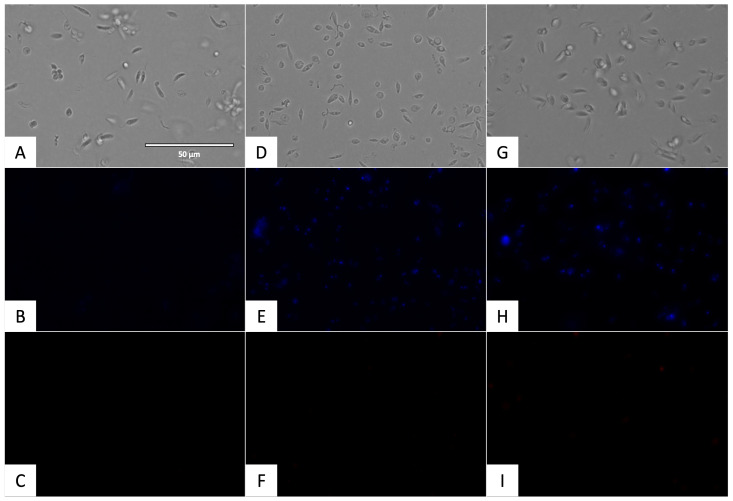
Detection of chromatin condensation using Hoechst/PI. Results after 24 h of incubation with the IC_90_ against promastigotes of *L. amazonensis.* Images were captured using an EVOS FL Cell Imaging System (40×). (**A**) Parasites without treatment in visible channel; (**B**) parasites without treatment in DAPI channel; (**C**) parasites without treatment in RFP channel; (**D**) parasites treated with flucofuron in visible channel; (**E**) parasites treated with flucofuron in DAPI channel; (**F**) parasites treated with flucofuron in RFP channel; (**G**) parasites treated with miltefosine in visible channel; (**H**) parasites treated with miltefosine in DAPI channel; and (**I**) parasites treated with miltefosine in RFP channel. Scale bar: 50 μm.

**Figure 8 pharmaceuticals-17-00554-f008:**
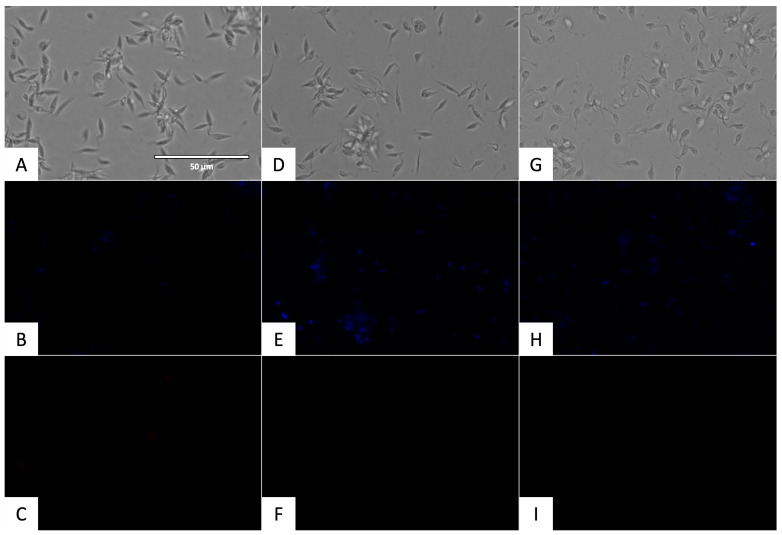
Detection of chromatin condensation using Hoechst/PI. Results after 24 h of incubation with the IC_90_ against epmastigotes of *T. cruzi.* Images were captured using an EVOS FL Cell Imaging System (40×). (**A**) Parasites without treatment in visible channel; (**B**) parasites without treatment in DAPI channel; (**C**) parasites without treatment in RFP channel; (**D**) parasites treated with flucofuron in visible channel; (**E**) parasites treated with flucofuron in DAPI channel; (**F**) parasites treated with flucofuron in RFP channel; (**G**) parasites treated with benznidazole in visible channel; (**H**) parasites treated with benznidazole in DAPI channel; and (**I**) parasites treated with benznidazole in RFP channel. Scale bar: 50 μm.

**Figure 9 pharmaceuticals-17-00554-f009:**
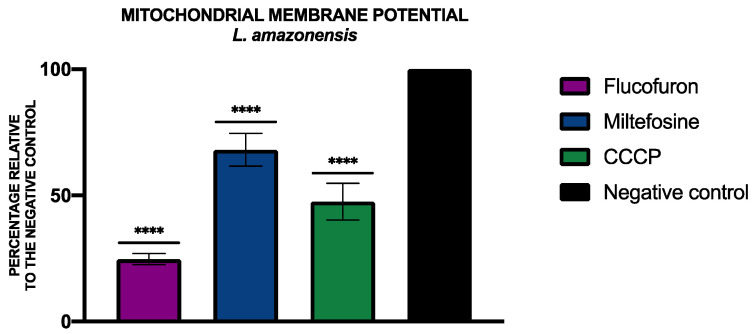
Results of mitochondrial membrane potential alterations expressed as percentages relative to negative control. Miltefosine was added as the standard treatment, and carbonyl cyanide m-chlorophenyl hydrazone (CCCP) was used as positive control. A Tukey test with GraphPad.PRISM^®^ 9.0.0 software was used to test the statistical differences between means. (**** *p* < 0.0001).

**Figure 10 pharmaceuticals-17-00554-f010:**
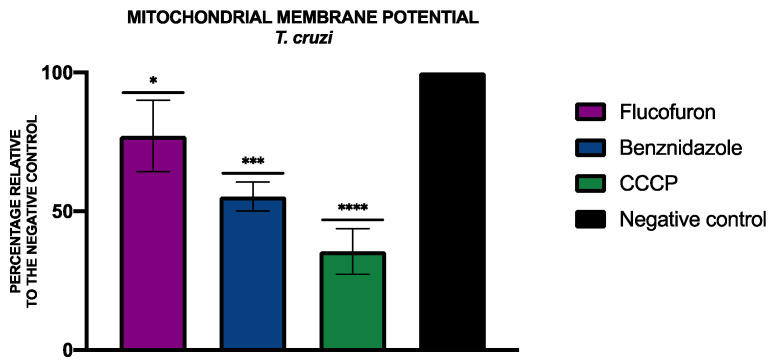
Results of mitochondrial membrane potential alterations expressed as percentages relative to negative control. Benznidazole was added as the standard treatment, and carbonyl cyanide m-chlorophenyl hydrazone (CCCP) was used as positive control. A Tukey test with GraphPad.PRISM^®^ 9.0.0 software was used to test the statistical differences between means. (* *p* < 0.05; *** *p* < 0.001; **** *p* < 0.0001).

**Figure 11 pharmaceuticals-17-00554-f011:**
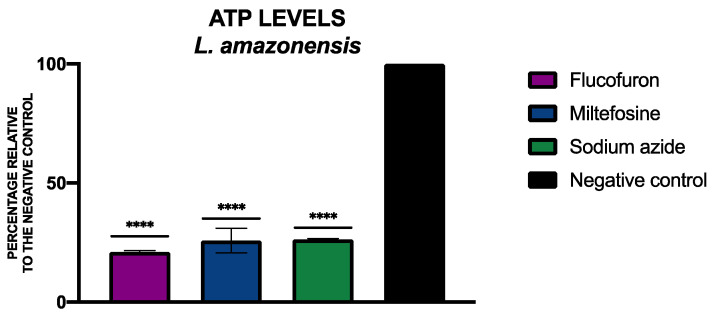
Results of ATP levels represented as the percentages relative to negative control. Miltefosine was used as the standard treatment, and sodium azide was used as positive control. A Tukey test with GraphPad.PRISM^®^ 9.0.0 software was used to test the statistical differences between means (**** *p* < 0.0001).

**Figure 12 pharmaceuticals-17-00554-f012:**
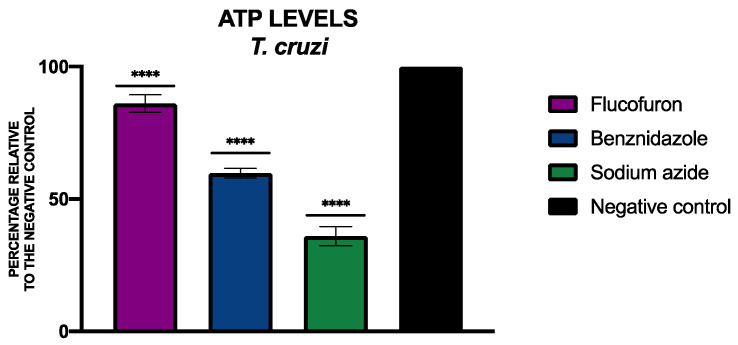
Results of ATP levels represented as the percentages relative to negative control. Benznidazole was used as the standard treatment, and sodium azide was used as positive control. A Tukey test with GraphPad.PRISM^®^ 9.0.0 software was used to test the statistical differences between means (**** *p* < 0.0001).

**Figure 13 pharmaceuticals-17-00554-f013:**
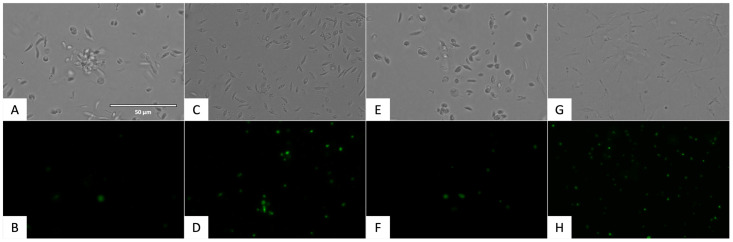
Detection of plasmatic membrane permeability in promastigote stage of *L. amazonensis* using SYTOX^®^ Green staining (ThermoFisher Scientific, Waltham, MA, USA). Images were captured using an EVOS FL Cell Imaging System (40×). Scale bar: 50 µm. (**A**) Untreated parasites in visible channel; (**B**) untreated parasites in GFP channel; (**C**) parasites treated with flucofuron in visible channel; (**D**) parasites treated with flucofuron in GFP channel; (**E**) parasites treated with miltefosine, standard treatment, in visible channel; (**F**) parasites treated with miltefosine, standard treatment, in GFP channel; (**G**) parasites treated with Triton 0.1%, positive control, in visible channel; and (**H**) parasites treated with Triton 0.1%, positive control, in GFP channel.

**Figure 14 pharmaceuticals-17-00554-f014:**
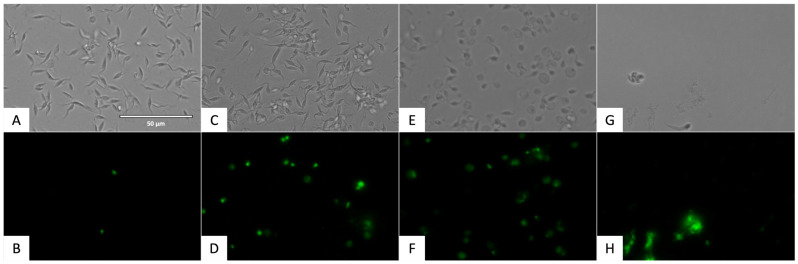
Detection of plasmatic membrane permeability in epimastigote stage of *T. cruzi* using SYTOX^®^ Green staining (ThermoFisher Scientific, Waltham, MA, USA). Images were captured using an EVOS FL Cell Imaging System (40×). Scale bar: 50 µm. (**A**) Untreated parasites in visible channel; (**B**) untreated parasites in GFP channel; (**C**) parasites treated with flucofuron in visible channel; (**D**) parasites treated with flucofuron in GFP channel; (**E**) parasites treated with benznidazole, standard treatment, in visible channel; (**F**) parasites treated with benznidazole, standard treatment, in GFP channel; (**G**) parasites treated with Triton 0.1%, positive control, in visible channel; and (**H**) parasites treated with Triton 0.1%, positive control, in GFP channel.

**Figure 15 pharmaceuticals-17-00554-f015:**
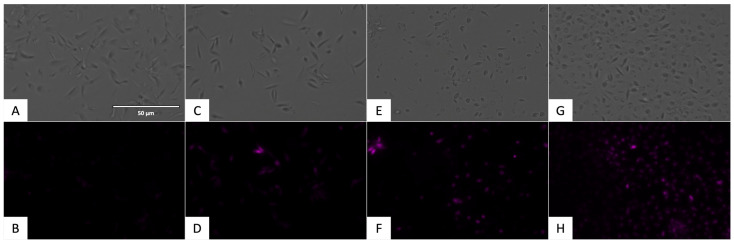
Detection of reactive oxygen species in promastigote stage of *L. amazonensis* using CellROX^®^ Deep Red staining (ThermoFisher Scientific, Waltham, MA, USA). Images were captured using an EVOS FL Cell Imaging System (40×). Scale bar: 50 µm. (**A**) Untreated parasites in visible channel; (**B**) untreated parasites in Cy5 channel; (**C**) parasites treated with miltefosine, standard treatment, in visible channel; (**D**) parasites treated with miltefosine, standard treatment, in Cy5 channel; (**E**) parasites treated with flucofuron in visible channel; (**F**) parasites treated with flucofuron in Cy5 channel; (**G**) parasites treated with H_2_O_2_ 600 mM, positive control, in visible channel; and (**H**) parasites treated with H_2_O_2_ 600 mM, positive control, in Cy5 channel.

**Figure 16 pharmaceuticals-17-00554-f016:**
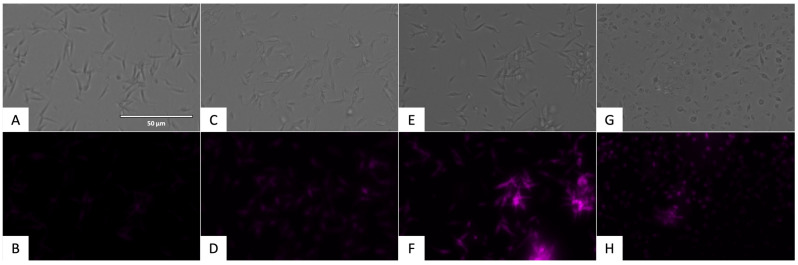
Detection of reactive oxygen species in epimastigote stage of *T. cruzi* using CellROX^®^ Deep Red staining (ThermoFisher Scientific, Waltham, MA, USA). Images were captured using an EVOS FL Cell Imaging System (40×). Scale bar: 50 µm. (**A**) Untreated parasites in visible channel; (**B**) untreated parasites in Cy5 channel; (**C**) parasites treated with benznidazole, standard treatment, in visible channel; (**D**) parasites treated with benznidazole, standard treatment, in Cy5 channel; (**E**) parasites treated with flucofuron in visible channel; (**F**) parasites treated with flucofuron in Cy5 channel; (**G**) parasites treated with H_2_O_2_ 600 mM, positive control, in visible channel; and (**H**) parasites treated with H_2_O_2_ 600 mM, positive control, in Cy5 channel.

**Figure 17 pharmaceuticals-17-00554-f017:**
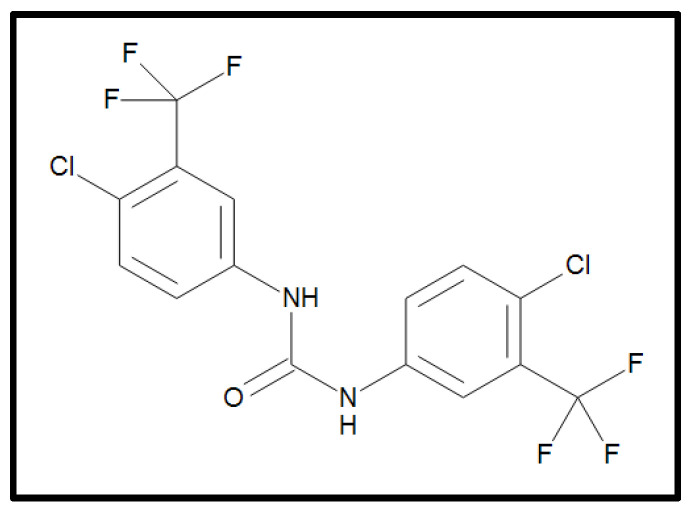
Molecular structure of flucofuron. The structure of flucofuron is C_15_ H_8_ Cl_2_ F_6_ N_2_ O (MW = 417.13 g/mol). The image was obtained from LGC Standards, Dr. Ehrenstorfer^TM^ (LGC group, Barcelona, Spain).

**Table 1 pharmaceuticals-17-00554-t001:** Results of in vitro activity against intracellular and extracellular forms of *L. amazonensis* and *T. cruzi* and cytotoxicity against murine macrophages. The results of IC_50_ and CC_50_ were expressed in mean ± standard deviation.

	CC_50_ Murine Macrophages (µM)	IC_50_ *Leishmania amazonensis* (µM)		SI
**Flucofuron**	83.86 ± 20.76	Promastigote	6.07 ± 1.11	13.8
		Amastigote	3.14 ± 0.39	26.7
**Miltefosine**	72.18 ± 8.85	Promastigote	6.48 ± 0.25	11.1
		Amastigote	3.12 ± 0.30	23.1
	**CC_50_ Murine Macrophages (µM)**	**IC_50_ *Trypanosoma cruzi* (µM)**		**SI**
**Flucofuron**	83.86 ± 20.76	Epimastigote	4.28 ± 0.83	19.6
		Amastigote	3.26 ± 0.34	25.7
**Benznidazole**	399.91 ± 1.40	Epimastigote	6.92 ± 0.77	57.8
		Amastigote	2.67 ± 0.39	149.8

IC_50_: inhibitory concentration 50; CC_50_: cytotoxic concentration 50; SI: selectivity index (CC_50_/IC_50_).

## Data Availability

Data are contained within the article and Appendix A.

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
