# Peer review of "Global Health Priority Box: Discovering Flucofuron as a Promising Antikinetoplastid Compound"

_pharmaceuticals, 2024, doi:10.3390/ph17050554_

Round 1

Reviewer 1 Report

Comments and Suggestions for Authors

Dear Authors,

After carefully reviewing your manuscript, while the content is intriguing, there is room for improvement in the clarity of the language, particularly in evaluating a library of compounds against two parasites responsible for neglected diseases.

One notable concern is the need for a sequential order in presenting the figures. Example: Figure 1 requires a more detailed description because it is not described in the manuscript.

Furthermore, the legends accompanying the figures need more crucial information, such as details on the evaluated elements, the statistical methods applied, and the controls utilized.

The images obtained from the EVOS FL Cell Imaging System pose a challenge due to insufficient resolution. While brightfield images appear satisfactory, the immunofluorescence images exhibit inferior resolution, making it difficult to discern if they contain artifacts. Images with high-resolution or confocal microscopy are needed.

The manuscript mentions a higher percentage of activity for the compounds, but unfortunately, crucial details such as IC50 values are omitted. 

Additionally, IC50 graphs must be present, hindering a comprehensive understanding of the observed effects and their reproducibility. Furthermore, details regarding biological and experimental triplicates in the conducted assays need to be included.

Author Response

REVIEWER 1:

Dear Authors,

After carefully reviewing your manuscript, while the content is intriguing, there is room for improvement in the clarity of the language, particularly in evaluating a library of compounds against two parasites responsible for neglected diseases.

Thank you very much for your positive feedback. Please find below detailed point by point responses to your comments along with the references to the manuscript. The changes implemented have been highlighted in yellow.

One notable concern is the need for a sequential order in presenting the figures. Example: Figure 1 requires a more detailed description because it is not described in the manuscript.
Furthermore, the legends accompanying the figures need more crucial information, such as details on the evaluated elements, the statistical methods applied, and the controls utilized.

Thank you for the comment. As you mentioned, the figures had a sequencing error, so figure 1 was not described in the manuscript. The sequencing error has been corrected, and more information on methods and controls has been added in the figure captions. Some figures do not apply statistical methods, they are descriptive rather than analytical statistics.

The images obtained from the EVOS FL Cell Imaging System pose a challenge due to insufficient resolution. While brightfield images appear satisfactory, the immunofluorescence images exhibit inferior resolution, making it difficult to discern if they contain artifacts. Images with high-resolution or confocal microscopy are needed.

Thank you for the comment. As you comment, with confocal microscopy could obtain images with better resolution, but we are trying to only see if there are higher number of parasites with fluorescence. With the EVOS FL Cell Imaging System we can obtain a qualitative result with which we can conclude the PCD produced in the parasites.  

The manuscript mentions a higher percentage of activity for the compounds, but unfortunately, crucial details such as IC50 values are omitted. 

Thank you for the comment. As you commented, only the most active compound (flucofuron) was used to calculate the IC50. We use the percentage of inhibition of the parasites to select the best product (as screening of the box), with these results we selected the flucofuron and purchase it to carry out the rest of the assays.

Additionally, IC50 graphs must be present, hindering a comprehensive understanding of the observed effects and their reproducibility. Furthermore, details regarding biological and experimental triplicates in the conducted assays need to be included.

Thank you for the comment. As you recommended, the IC50 graphs were added as supplementary material (Figure S1). Furthermore, sorry about the lack of information about biological and experimental triplicates in the conducted assays, the information was added.

Reviewer 2 Report

Comments and Suggestions for Authors

This study aims to screen the  Global Health Priority Box from MMV to identify new antileshmanial candidates.

Discovery new scaffold to fight this neglicted tropical disease is a matter of interest.

Some modifications before publication :

Italicize name of parasites (abstract and intro)

In keywords : what is the signification for PCD ?

For figure 1, please precise the signification for ZND, MB2 and VEC. It was only found after in the manuscript.

line 90 : figure 5 (not the 7)

In table 1, uniuts are missing as well as selective index SI as title

In the discussion part, the part relative to the mechanism of action should be be enriched.

A few word would be provided about the potential of flucofuron, i.e. ADME properties (even predicted), and provide a few avenues for its improvement.

Author Response

REVIEWER 2:

This study aims to screen the  Global Health Priority Box from MMV to identify new antileshmanial candidates.

Discovery new scaffold to fight this neglicted tropical disease is a matter of interest.

Thank you very much for your positive feedback. Please find below detailed point by point responses to your comments along with the references to the manuscript. The changes implemented have been highlighted in yellow.

Some modifications before publication :

Italicize name of parasites (abstract and intro)

Thank you for the comment. The names of the parasites were changed to italics.

In keywords : what is the signification for PCD ?

Thank you for the comment. With PCD we trying to talk about Programmed Cell Death. As your recommendation we change the keyword.

For figure 1, please precise the signification for ZND, MB2 and VEC. It was only found after in the manuscript.

Thank you for the comment. The signification was added to the legend.

line 90 : figure 5 (not the 7)

Thank you for your comment. The sequencing error has been corrected.

In table 1, uniuts are missing as well as selective index SI as title

Thank you for the comment. As you recommended the tittles and units were added. 

In the discussion part, the part relative to the mechanism of action should be be enriched.

Thank you for the comment. As you recommended, the discussion about the mechanisms of action was extended.

A few word would be provided about the potential of flucofuron, i.e. ADME properties (even predicted), and provide a few avenues for its improvement.

Thank you for the comment. As you recommended, a in silico prediction was added to discussion part.

Reviewer 3 Report

Comments and Suggestions for Authors

This is a very well planned and executed project dealing with the test of a large number of compounds that are part of the Global Health Priority Box. One compound, known as FLUCOFURAN, showed good activity against promastigote and amastigote forms of Leishmania amazonensis and against epimastigote and amastigote forms of Trypanosoma cruzi. IC50 lower tan 4 micromolar was observed against both parasites. The authors analyzed parameters such as presence of reactive oxygen species, integrity of the membrane permeability, cytotoxicity against macrophages, cell death process, mitochondria membrane potential and ATP levels. I would suggest an improvement of conclusion sumarizing the results obtained using the various approaches and their implication for the mechanim of antiparasitic activity of flucofuran. 

Author Response

REVIEWER 3:

This is a very well planned and executed project dealing with the test of a large number of compounds that are part of the Global Health Priority Box. One compound, known as FLUCOFURAN, showed good activity against promastigote and amastigote forms of Leishmania amazonensis and against epimastigote and amastigote forms of Trypanosoma cruzi. IC50 lower tan 4 micromolar was observed against both parasites. The authors analyzed parameters such as presence of reactive oxygen species, integrity of the membrane permeability, cytotoxicity against macrophages, cell death process, mitochondria membrane potential and ATP levels. I would suggest an improvement of conclusion sumarizing the results obtained using the various approaches and their implication for the mechanim of antiparasitic activity of flucofuran.

Thank you very much for your positive feedback. As you recommended the conclusion was extended. The changes have been highlighted in yellow.

Reviewer 4 Report

Comments and Suggestions for Authors

In this research, 240 compounds that are part of the Global Health Priority Box were used to test their activity against the protozoan parasites L. amazonensis and Trypanosoma cruzi and were also used against murine macrophages as control. One of these compounds, flucofuron demonstrates a higher percentage of inhibition and shows activity against both microorganisms producing programmed cell death. 

In general, the work is novel and interesting for the scientific community on the way to the search for therapeutic alternatives for leishmaniasis and Chagas disease. 

Some points need to be corrected. 

Major corrections

>In my opinion, the title does not reflect the work that was done. It seems to me that if only one of the compounds proved to have great anti-kinetoplastid activity, it cannot be concluded that all the compounds in the library have activity against these species and the title somewhat implies this. I think the title should be reconsidered for one that better reflects the work in general. 

>The introduction is a little short to explain the importance of the search for new drugs for these parasites, the mechanisms of action, and to give a better context of the library of compounds that were used, where do they come from? , the nature of compounds? , chemical structure? , etc.  

>In figures 1, 2, 5, and 6 the authors mention the acronyms ZND, MB2, and VEC without any context, so far the text has not been mentioned to which they refer and only mentioned until the section on materials and methods. Figure feet are not explanatory.

>Figures 3, 4, 5, and 6 seem to me unnecessary since they are redundant and do not provide new information, it is the same information already presented but graphed in another way.

>The numeric values in Table 1 are unclear because the table does not have the headers for each column. 

>The authors claim that the flucofuron compound induces the condensing of chromatin, however, Figures 7 and 8 are not clear and throughout the text, there is no clear mention of which are and why each of the channels were used in microscopy experiments. 

Minor corrections

> The names of microorganisms are not written in italics throughout the text

>Figure 17 does not mention whether it was created by the authors or copied from another source which requires proper citation.

Author Response

REVIEWER 4:

In this research, 240 compounds that are part of the Global Health Priority Box were used to test their activity against the protozoan parasites L. amazonensis and Trypanosoma cruzi and were also used against murine macrophages as control. One of these compounds, flucofuron demonstrates a higher percentage of inhibition and shows activity against both microorganisms producing programmed cell death. 

In general, the work is novel and interesting for the scientific community on the way to the search for therapeutic alternatives for leishmaniasis and Chagas disease. 

Some points need to be corrected. 

Thank you very much for your positive feedback. Please find below detailed point by point responses to your comments along with the references to the manuscript. The changes implemented have been highlighted in yellow.

Major corrections

>In my opinion, the title does not reflect the work that was done. It seems to me that if only one of the compounds proved to have great anti-kinetoplastid activity, it cannot be concluded that all the compounds in the library have activity against these species and the title somewhat implies this. I think the title should be reconsidered for one that better reflects the work in general. 

Thank you for the comment. As you recommended, the title was changed to: “Global Health Priority Box: Discovering Flucofuron as a Promising Antikinetoplastid Compound”.

>The introduction is a little short to explain the importance of the search for new drugs for these parasites, the mechanisms of action, and to give a better context of the library of compounds that were used, where do they come from? , the nature of compounds? , chemical structure? , etc.

Thank you for the comment. As you recommended, the introduction was extended paying attention in the importance of the development of new therapies against these pathologies. On the other hand, the molecular structure of the compounds was not added because it is available in the MMV web page (https://www.mmv.org/mmv-open/global-health-priority-box/about-global-health-priority-box), indicated in the introduction part. This information was added to materials and methods part, in the explanation of the products.

>In figures 1, 2, 5, and 6 the authors mention the acronyms ZND, MB2, and VEC without any context, so far the text has not been mentioned to which they refer and only mentioned until the section on materials and methods. Figure feet are not explanatory.

Thank you for the comment. The signification was added to the legend.

>Figures 3, 4, 5, and 6 seem to me unnecessary since they are redundant and do not provide new information, it is the same information already presented but graphed in another way.

Thank you for the comment. We think these graphs are explanatory and necessary to visually clarify the number of active compounds per parasite and compound type. The different mood to present these graphs try to make attention in the number of active compounds by parasite (graphs 3 and 4) and by plate/type of products (graphs 5 and 6). We are aware that these graphs do not present new information, but we think that for the reader it is more visible to understand the percentage of high active compounds.

>The numeric values in Table 1 are unclear because the table does not have the headers for each column.

Thank you for the comment. As you recommended the values and explanation were added. 

>The authors claim that the flucofuron compound induces the condensing of chromatin, however, Figures 7 and 8 are not clear and throughout the text, there is no clear mention of which are and why each of the channels were used in microscopy experiments. 

Thank you for the comment. Commercial kits were used for programmed cell death studies. The manufacturers of these kits report the corresponding wavelength ranges at which fluorescence will occur. These wavelengths are listed in material and methods for each kit and the corresponding channel used (the missing CellROX range has been added). On the other hand, throughout the text, both in material and methods and in results, it is detailed what each fluorescence indicates, for example blue indicates chromatin condensation when using the Hoechst kit, red indicates accumulation of reactive oxygen species when using CellROX or the presence of green with the use of SytoxGreen would indicate alterations in the permeability of the plasma membrane.

Minor corrections

> The names of microorganisms are not written in italics throughout the text

Thank you for the comment. The name of the parasites were changed to italics.

>Figure 17 does not mention whether it was created by the authors or copied from another source which requires proper citation.

Thank you for the comment. The flucofuron was adquired by LGC Standards, Dr. Ehrenstorfer™️ (LGC group, Barcelona, Spain). The image was obtained by the same company.

Reviewer 5 Report

Comments and Suggestions for Authors

The manuscript tests the activity of a library of compounds, already evaluated for Plasmodium sp, against Leishmania sp and T. cruzi aiming at drug repositioning. The work is new and important for the discovery of new therapeutic compounds. It is well presented; however, it needs updating on some points before being accepted for publication,

Major points

1) After the first description of Trynosma cruzi and Leishmania sp, the names must be abbreviated and in italics. They also need to be italicized at several points throughout the text.

2)Line 294, change “de” to “the.”

3)We had difficulty analyzing Figure 1, Figure 2, and Table 1. The representative points of the compounds in Figures 1 and 2 are very small, and the color difference does not stand out.

4) In Table 1, the footer (IC50: inhibitory concentration 50; CC50: cytotoxic concentration 50; SI: selectivity index) does not mean anything about what is being seen in the table.

5) Similarly, the activity values presented for macrophages must be better understood. This table must be modified to be more explicit to readers.

We think enlarging figures 7, 8, 13, and 16 is necessary. The activities of parasites are difficult to visualize. We understand it isn't easy due to the space available, but perhaps the authors could expand them a little more.

6)The discussion is not well presented and should be improved. Several publications can enrich the text, including using compounds extracted from plants with activity on both parasites. Furthermore, some sentences/words have already been presented in the Introduction, and their presentation should be changed.

7)Another fact not discussed is these compounds' solubility and relationship with the test used to evaluate the activity and possible in vivo activity.

8)The conclusions are weak and do not reflect what was observed in the results. Needs improvement.

9)The references are current and necessary for the composition of the text but can be expanded.

Comments on the Quality of English Language

English is fine.

Author Response

REVIEWER 5:

The manuscript tests the activity of a library of compounds, already evaluated for Plasmodium sp, against Leishmania sp and T. cruzi aiming at drug repositioning. The work is new and important for the discovery of new therapeutic compounds. It is well presented; however, it needs updating on some points before being accepted for publication,

Thank you very much for your positive feedback. As you recommended the conclusion was extended. The changes have been highlighted in yellow.

Major points

1) After the first description of Trynosma cruzi and Leishmania sp, the names must be abbreviated and in italics. They also need to be italicized at several points throughout the text.

Thank you for the comment. The name of the parasites were changed to italics.

2)Line 294, change “de” to “the.”

Thank you for the comment. The word was changed.

3)We had difficulty analyzing Figure 1, Figure 2, and Table 1. The representative points of the compounds in Figures 1 and 2 are very small, and the color difference does not stand out.

Thank you for the comment. The figures 1 and 2 were increased in size, the points could not be made bigger as they overlap with each other. The table 1 was modified trying to make it more understandable.

4) In Table 1, the footer (IC50: inhibitory concentration 50; CC50: cytotoxic concentration 50; SI: selectivity index) does not mean anything about what is being seen in the table.

Thank you for the comment. The table 1 was modified and the headers of each column were added.

5) Similarly, the activity values presented for macrophages must be better understood. This table must be modified to be more explicit to readers.

Thank you for the comment. The same information was added for the macrophages (CC50).

We think enlarging figures 7, 8, 13, and 16 is necessary. The activities of parasites are difficult to visualize. We understand it isn't easy due to the space available, but perhaps the authors could expand them a little more.

Thank you for the comment. The images were enlarged as much its possible.

6)The discussion is not well presented and should be improved. Several publications can enrich the text, including using compounds extracted from plants with activity on both parasites. Furthermore, some sentences/words have already been presented in the Introduction, and their presentation should be changed.

Thank you for the comment. The discussion was improved, we hope we have gathered the necessary information.

7)Another fact not discussed is these compounds' solubility and relationship with the test used to evaluate the activity and possible in vivo activity.

Thank you for the comment. The solubility of the compounds was good for DMSO, although we are aware that this solvent has more dissolving power than physiological liquids. A discussion of in silico data on flucofuron has been added. We do not currently have more information on its solubility for an in vivo assay, but we believe it may be a possible assay for future testing for an in vivo assay.

8)The conclusions are weak and do not reflect what was observed in the results. Needs improvement.

Thank you for the comment. The conclusions section was improved.

9)The references are current and necessary for the composition of the text but can be expanded.

Thank you for the comment. The references were extended according to the information added.

Round 2

Reviewer 1 Report

Comments and Suggestions for Authors

Dear Authors,

Thank you for taking the time to follow some of the suggestions. However, the new version of the manuscript still needs significant improvement. The observations are below.

Lane 31. Update the reference numbering. The manuscript shows that it usually starts with 1, not 2.

Lane 36. The WHO website describes 8 million people; please always use updated information.

Lanes 36-38. The information needs to be more accurate; we need to know which host is referenced. Is it on humans?

Lane 94. It is without treatment, but it has the vehicle buffer, right? (Is it DMSO plus other reagents? Please add the information on the materials and methods sections, described here as a “vehicle buffer.” Treatment without anything is not a negative control.

Please clarify the legend.

Legends in Figures 1 and 2 need more description; for example, does each dot mean? Which statistical method was used, and if 0, 50, or 100 are %? Do not assume we know the information.

Also, an arrowhead should be used to show the readers the molecule of interest (Flucofuron).

In Figures 3 and 4, which methodology generated those results? How were those methodologies performed? And what is the significance? Did the authors include a gold-standard methodology? Was it performed in experimental and biological replications?

In Figures 5 and 6, which methodology was followed? Which statistical methods were used? Is it a prism? Excel? Stata? In addition to the percentage, please add the number of compounds in each group and the kind of compounds in each group. Did they have similar smiles, structures, chemicals, and MOAs? All this information is included in any MMV box, so it should be analyzed.

Table 1. Please add a legend and describe your findings.

Figure 7. The figure has a poor-quality resolution: DAPI should stain the nuclei for each cell, a z-stack image should/could be performed to optimize the results, and the maximum project needs to be exported and shown. The actual images are not acceptable.

Figures 8 and 9 are of the same poor quality. Hoechst and DAPI are very sensitive dyes and should stain the nuclei for each cell; this data has poor resolution. The authors need advice from an expert in high-resolution images.

Figures 13 and 14 had the same problem. SYTO dye should stain nuclei from cells and parasites for each cell shown in the images, which is not the case. These images are of poor quality. The plane image may not be the most representative, and the authors need to run a z-stack and export a maximum project image.

Figures 15 and 16. These images might be accurate. However, the authors did not include any dye control, such as a specific antibody that recognizes the parasite explicitly. The signal looks like a background. 

Prism files for all 240 compounds are not included (summarized in 3 panels for each 80 group compounds); it was requested in the previous version.

The manuscript needs significant improvement and cannot be accepted in its present form.

Author Response

REVIEWER 1:

Dear Authors,

Thank you for taking the time to follow some of the suggestions. However, the new version of the manuscript still needs significant improvement. The observations are below.

Thank you very much for your positive feedback. Please find below detailed point by point responses to your comments along with the references to the manuscript. The changes implemented have been highlighted in yellow.

Lane 31. Update the reference numbering. The manuscript shows that it usually starts with 1, not 2.

Thank you for the comment. The bibliography was corrected.

Lane 36. The WHO website describes 8 million people; please always use updated information.

Thank you for the comment. As you recommended, the WHO website was visited https://www.who.int/news-room/fact-sheets/detail/chagas-disease-(american-trypanosomiasis)

. In its latest entry on Chagas disease, dated 6 April 2023, it indicates between 6 and 7 million people, which is why it added approximately 7. I don't know where you got the figure of 8 million, but it is impossible for me to find it.

Lanes 36-38. The information needs to be more accurate; we need to know which host is referenced. Is it on humans?

Thank you for the comment. The lines to which you refer, cited below, clearly speak of people, and we do not think that they are open to doubt.

“Chagas disease is estimated to affect approximately 7 million people, mainly in Latin America, and can lead to major chronic disease of the intestine and heart, which can kill up to 7000 people a year [3,4]. On the other hand, leishmaniasis affects more than 12 million people, causing, mainly in its visceral and most severe form, about 30 thousand deaths per year [5].”

Lane 94. It is without treatment, but it has the vehicle buffer, right? (Is it DMSO plus other reagents? Please add the information on the materials and methods sections, described here as a “vehicle buffer.” Treatment without anything is not a negative control.

Please clarify the legend.

Thank you for the comment. The negative control is only LIT culture medium, without treatment. It is not necessary to add DMSO, as previous controls have shown that concentrations of less than 2% DMSO do not affect parasites, so whenever treatment is added, it is always below this concentration. However, the legends and material and methods were modified as you recommend.

Legends in Figures 1 and 2 need more description; for example, does each dot mean? Which statistical method was used, and if 0, 50, or 100 are %? Do not assume we know the information.

Also, an arrowhead should be used to show the readers the molecule of interest (Flucofuron).

Thank you for the comment. The legends of the images 1 and 2 were modified, and the arrow for flucofuron were added. These figures are only a graphical representation of the activity, no statistical study has been carried out.

In Figures 3 and 4, which methodology generated those results? How were those methodologies performed? And what is the significance? Did the authors include a gold-standard methodology? Was it performed in experimental and biological replications?

In Figures 5 and 6, which methodology was followed? Which statistical methods were used? Is it a prism? Excel? Stata? In addition to the percentage, please add the number of compounds in each group and the kind of compounds in each group. Did they have similar smiles, structures, chemicals, and MOAs? All this information is included in any MMV box, so it should be analyzed.

Thank you for the comment. As in the case of figures 1 and 2, figures 3, 4, 5 and 6 are only a graphical representation of the results, they do not include any statistical study. Figures 3 and 4 represent the number of active compounds per parasite, grouped by activity groups. Images 5 and 6 do the same, but this time attention is paid to the kind of compound, MB2, ZND and VEC. The number of compounds of each group were added in images 3, 4, 5 and 6.

On the other hand, as you recommend, it would be interesting to do a study of the structure-activity of the compounds. We believe that the data present in this study are not sufficient to do so, since only the IC50 of one compound (flucofuron) has been studied, and this is not the focus of the present study. The structure data are accessible and public on the MMV website, so we do not believe that they should be added in this work.

Table 1. Please add a legend and describe your findings.

Thank you for the comment. A paragraph explaining the table was added.

Figure 7. The figure has a poor-quality resolution: DAPI should stain the nuclei for each cell, a z-stack image should/could be performed to optimize the results, and the maximum project needs to be exported and shown. The actual images are not acceptable.

Figures 8 and 9 are of the same poor quality. Hoechst and DAPI are very sensitive dyes and should stain the nuclei for each cell; this data has poor resolution. The authors need advice from an expert in high-resolution images.

Figures 13 and 14 had the same problem. SYTO dye should stain nuclei from cells and parasites for each cell shown in the images, which is not the case. These images are of poor quality. The plane image may not be the most representative, and the authors need to run a z-stack and export a maximum project image.

Figures 15 and 16. These images might be accurate. However, the authors did not include any dye control, such as a specific antibody that recognizes the parasite explicitly. The signal looks like a background. 

Thank you for the comment. The figures 8, 9, 13, 14, 15 and 16 were changed and enhanced with the GIMP program, containing a quality of 300 dpi, which is what is usually considered as a good resolution image.

On the other hand, in the case of the SYTOX images, it can be seen that in many of them the nucleus is stained, and sometimes even the kinetoplast. In certain cells where cell death is quite advanced, a more distributed fluorescence is observed throughout the cell, which refers to the most degraded genetic material.

In the cellROX images, the positive control added was H2O2, which has already been shown to be ROS-producing. When you refer to the fact that the images look like background, you have to take into account that the fluorescence coincides perfectly with the cellular interior of the visible images, in addition to the fact that it does not appear in the negative control and appears to a greater extent in the positive control, which does not correspond to a background, but to the fluorescence of the ROS production itself.

Finally, you referred to the need to perform a z-stack for these protocols. From our point of view and demonstrable experience with fluorescence imaging in protozoa, we do not see the need for z-stack to be able to draw the conclusions that we have drawn with this fluorescence study. Furthermore, we must take into account that the assays we have performed do not have fixation, the cells are in motion, and that many of these kits are incompatible with the fixation techniques necessary to perform a z-stack.

Prism files for all 240 compounds are not included (summarized in 3 panels for each 80 group compounds); it was requested in the previous version.

Thank you for the comment. As mentioned in material and methods, in addition to each graph specifically, the first screening is represented as percentage of live cells compared to the negative control. The IC50 is not calculated (only calculated in flucofuron), so only the prism files for flucofuron were added in the first version.

The manuscript needs significant improvement and cannot be accepted in its present form.

Reviewer 5 Report

Comments and Suggestions for Authors

Most of the observations were incorporated into the text appropriately, we consider that the manuscript should now be accepted for publication.

Comments on the Quality of English Language

English is fine, only small edits of a few words are necessary.

Author Response

REVIEWER 5:

Most of the observations were incorporated into the text appropriately, we consider that the manuscript should now be accepted for publication.

English is fine, only small edits of a few words are necessary.

Thank you very much for your positive feedback.
